The whole-genome and expression profile analysis of WRKY and RGAs in Dactylis glomerata showed that DG6C02319.1 and DgWRKYs may cooperate in the immunity against rust

Ren Juncai 1
Hu Jialing 2
Zhang Ailing 2
Ren Shuping 1
Jing Tingting 2
Wang Xiaoshan 2
Sun Min 2
http://orcid.org/0000-0001-7810-4852 Huang Linkai 2 huanglinkai@sicau.edu.cn
Zeng Bing 1 zbin78@163.com
1 College of Animal Science and Technology, Southwest University , Chongqing, Chongqing , China
2 College of Grassland Science and Technology, Sichuan Agricultural University , Chengdu, Sichuan , China
Engelberth Jurgen
Electronic publication date: 2021 Aug 19
Publication date: 2021
Volume: 9
Electronic Location ID: e11919
Received 2021 Feb 15; Accepted 2021 Jul 16
Copyright: © 2021 Ren et al.
Copyright year: 2021
Copyright holder: Ren et al.
License: This is an open access article distributed under the terms of the Creative Commons Attribution License, which permits unrestricted use, distribution, reproduction and adaptation in any medium and for any purpose provided that it is properly attributed. For attribution, the original author(s), title, publication source (PeerJ) and either DOI or URL of the article must be cited.
License URL: https://creativecommons.org/licenses/by/4.0/

Keywords: Dactylis glomerate, WRKY family, RGAs, Abiotic stress, Rust stress, Transcriptome

Funding: National Natural Science Foundation of China 32071867 and 31771866 Chongqing Modern Agricultural Industry Technology System Program for Herbivore 2021(11) Fundamental Research Funds for the Central University XDJK2020RC001 This work was supported by the National Natural Science Foundation of China (Nos. 31771866 and 32071867), the Chongqing’s Modern Agricultural Industry Technology System Program for Herbivore (2021[12]) and the Fundamental Research Funds for the Central University (No. XDJK2020RC001). The funders had no role in study design, data collection and analysis, decision to publish, or preparation of the manuscript.

==============================
Orchardgrass (Dactylis glomerata) is one of the top four perennial forages worldwide and, despite its large economic advantages, often threatened by various environmental stresses. WRKY transcription factors (TFs) can regulate a variety of plant processes, widely participate in plant responses to biotic and abiotic stresses, and are one of the largest gene families in plants. WRKYs can usually bind W-box elements specifically. In this study, we identified a total of 93 DgWRKY genes and 281 RGAs, including 65, 169 and 47 nucleotide-binding site-leucine-rich repeats (NBS-LRRs), leucine-rich repeats receptor-like protein kinases (LRR-RLKs), and leucine-rich repeats receptor-like proteins (LRR-RLPs), respectively. Through analyzing the expression of DgWRKY genes in orchardgrass under different environmental stresses, it was found that many DgWRKY genes were differentially expressed under heat, drought, submergence, and rust stress. In particular, it was found that the greatest number of genes were differentially expressed under rust infection. Consistently, GO and KEGG enrichment analysis of all genes showed that 78 DgWRKY TFs were identified in the plant–pathogen interaction pathway, with 59 of them differentially expressed. Through cis-acting element prediction, 154 RGAs were found to contain W-box elements. Among them, DG6C02319.1 (a member of the LRR-RLK family) was identified as likely to interact with 14 DGWRKYs. Moreover, their expression levels in susceptible plants after rust inoculation were first up-regulated and then down-regulated, while those in the resistant plants were always up-regulated. In general, DgWRKYs responded to both biotic stress and abiotic stress. DgWRKYs and RGAs may synergistically respond to the response of orchardgrass to rust. This study provides meaningful insight into the molecular mechanisms of WRKY proteins in orchardgrass.

Introduction

Animal husbandry is one of the most important industries involved in feeding humans, and it can provide humans with a higher quality of life. The processing, production, and quality of forage grass directly influence the output of livestock production. Currently, many countries, especially developed countries, pay much attention to raising and studying forage grass because of its crucial roles in the economy as well as in nutrition. It has been estimated that forage grass occupies 26% of land area and 70% of agricultural land (Conant, 2010).

Orchardgrass (Dactylis glomerata L.), which belongs to the family Poaceae, is a cool-season forage grass planted worldwide (Hirata, Yuyama & Cai, 2011; Xie et al., 2014). As one of the top four perennial forage grasses from an economic perspective, orchardgrass is of vital importance in the production of meat from livestock and dairy production in temperate regions of the world (Wilkins & Humphreys, 2003). Furthermore, this species of grass possesses many advantages that have promoted its large-scale cultivation in America for grazing and hay harvest, including its fast growth, high yield, high sugar content, and shade tolerance (Tronsmo, 1993; Volaire, Conéjero & Lelièvre, 2001; Volaire, 2003). Additionally, it is also used to establish grasslands in some places (Brummer & Moore, 2000). Orchardgrass has a wide adaptative range, but pathogens and some abiotic stresses considerably affect its quality and yield. Orchardgrass is susceptible to infection by pathogens such as rust fungus, which has caused substantial damage to populations of orchardgrass in natural grasslands, resulting in an approximately 69% increase in yellow and brown leaves within rust-infected areas, as shown by Pfender & Alderman (2006). Similarly, Lancashire & Latch (2012) found that rust stress reduces the tiller number and biomass of orchardgrass. Abiotic stresses resulting from global climate change, such as drought, heat, and waterlogging, also threaten the growth and quality of orchardgrass (Huang et al., 2014; Huang et al., 2015; Majidi et al., 2014; Zandalinas et al., 2018). Accordingly, improving the tolerance of orchardgrass to biotic and abiotic stresses is essential, and a very important step in achieving this goal is mining genes associated with resistance.

WRKY is one of the most important gene families in plants and was named based on its conserved DNA binding domain sequence, WRKYGQK. This conserved domain is approximately 60 residues followed by a C2H2 (Cys2His2) or C2HC (Cys2HisCys) zinc-binding motif (Eulgem et al., 2000; Rushton et al., 1996). The WRKY family can be divided into three groups (I–III) according to the number of domains and the zinc finger structure. Group II WRKY proteins can be classified into a, b, c, d, and e subgroups based on their primary amino acid sequences (Wu et al., 2019; Yan et al., 2019; Zhang & Wang, 2005).

WRKY transcription factor (TF) family were found to have an important relationship with abiotic tolerance and plant growth in Teak (Tectona Grandis) (Wang et al., 2020a), Camellia japonica (Yang et al., 2020), Eucalyptus globulus (Aguayo et al., 2019), sesame (Sesamum indicum L.) (Wang et al. 2020b), peony (Paeonia lactiflora) (Wang et al., 2019), cassava (Manihot esculenta Crantz) (Wei et al., 2019) and so on. In cassava, mewrky20 can be activated by mehsp90.9 directly to encode a key enzyme involved in abscisic acid biosynthesis, and mewrky20-silenced plants displayed drought sensitivity, indicating its importance to the drought stress response (Wei et al., 2019). In pepper, overexpression of CaWRKY40 can enhance resistance to Ralstonia solanacearum and tolerance to heat stress of tobacco (Nicotiana tabacum L), while silencing of CaWRKY40 can enhance sensitivity to R. solanacearum and impair thermotolerance. Plwrky70 from P. lactiflora belongs to the group III WRKY family, which was considerably suppressed under waterlogging treatment, dramatically dropping to minimum levels after 2 h. This suggested that plwrky70 was sensitive to waterlogging stresses in P. lactiflora (Han et al., 2019). Other than that, WRKY proteins could play an important role in resistance to pathogen attack in wild potato (Solanum commersonii and S. chacoense), through their involvement in specific signaling pathways (Villano et al., 2020). Researchers isolated WRKY genes in rice (Oryza sativa L.) infected by the fungal pathogen Magnaporthe grisea, and 15 of 45 genes showed remarkably increased expression under infection. WRKY W109669 was able to induce tobacco endo-1,3-β-glucanase (NtPR2) and promote systemic defense responses to tobacco mosaic virus in transgenic tobacco plants (Naoumkina, He & Dixon, 2008). These results collectively indicate that many WRKY genes are very crucial to plant growth and plant resistance to abiotic stresses. Moreover, WRKY can participate in the process of plant defense against biological stress by regulating the expression of resistance genes (R genes) through several pathways.

Plants have complex pathogen recognition and defense mechanisms, including pathogen-associated molecular pattern-triggered immunity (PTI) and effect-triggered immunity (ETI), and R genes, play an important role in the process of disease resistance (Zhang et al., 2014). R genes have some common characteristics, and we can more broadly refer to resistance gene analogs (RGAs), which are genes with the structural characteristics of R genes. RGAs can be divided into nucleotide-binding site-leucine-rich repeats (NBS-LRRs) and transmembrane leucine-rich repeats (TM-LRRs) (Sekhwal et al., 2015). TM-LRRs can be further subdivided into two categories: leucine-rich repeats receptor-like protein kinases (LRR-RLKs) and leucine-rich repeats receptor-like proteins (LRR-RLPs) (Hammond-Kosack & Jones, 1997).

To elucidate the roles of WRKYs in different species and improve the current understanding of biotic and abiotic stress responses at the molecular level, the identification and analysis of WRKY family members in target plant species is essential. Studies on the WRKY TF family in rice (Ross, Liu & Shen, 2007; Ryu et al., 2006), Glycyrrhiza glabra (Goyal et al., 2020), Rosa chinensis (Liu et al., 2019), and Saccharum spontaneum (Li et al., 2019) have already been conducted. These studies reveal that the WRKY TF family is crucially involved in biotic and abiotic stress responses and that some of these genes have been produced by duplication events. However, there is still no published research on the WRKY family in orchardgrass. As such, this study aimed to identify and analyze the WRKY TF family in orchardgrass to provide a foundation for future molecular genetic improvement. Our research team has previously created high-quality expression profile data of orchardgrass under drought, heat, and submergence treatments (Huang et al., 2015; Ji et al., 2018; Zeng et al., 2020). These data enable clarification of the mechanism by which WRKY TFs function in D. glomerata. By analyzing these data, we found that most WRKY genes, relative to control conditions, were differentially expressed under biological and abiotic stresses in orchardgrass, especially under rust stress, with 80% of WRKY genes showing changes in expression level. To further analyze the regulatory mechanism by which DgWRKYs respond to rust, we identified all RGAs and their cis-acting elements and performed a weighted gene co-expression network analysis (WGCNA), revealing that DgWRKYs and RGAs were highly likely to interact.

Materials & Methods

Sequence retrieval

The published orchardgrass genome and protein sequences were downloaded from the orchardgrass genome database (http://orchardgrassgenome.sicau.edu.cn/download.php). The orchardgrass genome has been deposited under BioProject accession number PRJNA471014. The whole-genome assembly is composed of an approximately 1.84-Gb chromosome-scale diploid orchardgrass genome, including 40,088 protein-coding genes (Huang et al., 2020). The WRKY sequence data from A. thaliana (Araport11) were obtained from TAIR (https://www.arabidopsis.org/), while data from Triticum aestivum (using IWGSC(v2.2) gene annotation) were obtained from PlantTFDB (http://planttfdb.gao-lab.org/).

Identification of WRKY proteins from orchardgrass

We used two strategies to identify WRKY genes in D. glomerata. The first used HMMER SEARCH, in which we utilized HMMER v3 software (http://hmmer.janelia.org) to build an orchardgrass protein dataset. The Hidden Markov Model (HMM) file for WRKY (PF03106) domains was downloaded from Pfam (http://pfam.xfam.org/) (Finn et al., 2016) in order to identify WRKY proteins from the local database. The identification method (Lozano et al., 2015) was used to identify proteins using the raw WRKY HMM. A high-quality protein set (obtained using an E-value < 1 × 10−20 and manual verification of intact WRKY domains) was aligned and then used to construct an orchardgrass-specific WRKY HMM using hmmbuild from the HMMER v3 suite. Next, we used the new orchardgrass-specific HMM to scan the protein data, and all proteins with an E-value lower than 0.01 were selected.

The second method utilized BLASTP. First, we selected T. aestivum and O. sativa WRKYs (as shown in Table S1; all sequences were downloaded from NCBI https://www.ncbi.nlm.nih.gov/) as the query sequences for a BLAST search of the protein sequences of D. glomerata. The sequences with an E-value less than 1e−10 were selected for further analysis. Finally, all DgWRKY sequences were verified using the online tool Search Pfam (http://pfam.xfam.org/search/), while sequences without a WRKY domain were removed. The selected protein sequences are shown in Table S2.

Phylogenetic analysis and multiple sequence alignment

Phylogenetic trees of genes from T. aestivum, D. glomerata, and A. thaliana were constructed with MEGA X utilizing the maximum likelihood method with a Poisson correction model and 1,000 bootstrap replicates (Kumar et al., 2018). DNAMAN9 was used to analyze the core sequence of the WRKY domain from each subgroup of 93 DgWRKYs after multiple sequence alignment.

Chromosomal locations, motif analysis and gene structure of DgWRKYs

Chromosomal mapping was conducted using the MG2C online tool (http://mg2c.iask.in/mg2c_v2.1/). MEME (http://meme-suite.org/tools/meme/) was used to analyze motifs; the site distribution was any number of repetitions (ANR), and 10 consensus motifs were selected. Finally, the motifs and gene structure of DgWRKYs were mapped using TBtools. The coding sequence (CDS) of 93 orchardgrass genes were obtained from the orchardgrass genome data.

Protein physical and chemical properties analysis and subcellular localization prediction

The physicochemical properties of DgWRKYs were analyzed using ProtParam (https://web.expasy.org/protparam/), and estimates of the amino acid length, molecular weight (MW), theoretical isoelectric point (pI), instability index, aliphatic index, and grand average of hydropathicity (GRAVY) were obtained. Subcellular localization was predicted using WoLF PSORT (https://wolfpsort.hgc.jp/).

Gene duplications and Ka/Ks calculation

McscanX was used to perform a collinear analysis of the orchardgrass genome (Wang et al., 2012). BLASTN was used to perform homologous CDS sequence comparison of WRKY family members. Based on previous research, gene duplication was constrained to gene pairs with lengths of aligned CDSs greater than 75% of the longer sequence with a similarity of the aligned region greater than 75% (Gu et al., 2002). Then, if a pair of duplicates was on the same chromosome and there are fewer than five genes between the two given genes, they were considered tandem duplicates; otherwise, the genes were considered to be segmented repeats (Cheng et al., 2018).

A circos diagram was drawn using circos software (http://circos.ca/). KaKs_Calculator2.0 was used to calculate the nonsynonymous substitution rate, Ka, and the synonymous substitution rate, Ks, of each duplicate gene pair.

Expression profile analysis of WRKY genes under abiotic stresses and across different tissues

The expression pattern data of DgWRKY genes under different abiotic stresses and in varied tissues has been previously measured by our research team (Huang et al., 2020; Huang et al., 2015; Ji et al., 2018; Zeng et al., 2020). This research has included expression profiles of orchardgrass under heat, drought, and submergence stress published by Huang et al. (2015), Ji et al. (2018), and Zeng et al. (2020), respectively. However, genome data for this species was not published until 2020, which required this previous work to be performed as unreferenced transcriptome analyses. In the present study, the raw data for heat and drought stress were re-downloaded in order to perform a reference analysis, and the expression profiles of DgWRKYs under different abiotic stresses were thus obtained. All fragments per kilobase of exon per million reads mapped (FPKM) estimated under stress are shown in Table S3.

Expression profile analysis of WRKY genes under rust stress

Two types of orchardgrass genetic lines, highly resistant PI251814 and highly susceptible PI292589 lines, were used in this study. Four pots of the above two lines were cultivated, including two pots with highly resistant plants and two pots with highly susceptible plants. After a set cultivation period (at 20 ± 5 °C), one pot containing highly resistant plants and one pot containing highly susceptible plants were inoculated (treatment group), respectively, while the other two pots received no treatment (control group). We used smearing to inoculate the plants with rust fungus. The spore pile was picked from the grass experiment base of the Ya’an campus of Sichuan Agricultural University.

The inoculated plants were placed in an incubator for dark treatment for 24 h (12 °C, total darkness, 100% relative humidity) and then cultured for 14 days with 16-h days (20 ± 2 °C, 100% relative humidity) and 8-h nights (15 ± 2 °C, total darkness, 100% relative humidity). Latent spots appeared but were not obvious by 4 days after inoculation, while spore piles appeared by 7 days after inoculation. By 14 days after inoculation, the spore piles were fully mature, and the inoculated leaves showed symptoms of withering. The leaves of all plants were sampled on the 7th and 14th days for RNA-seq analysis (with two to three replicates per sample). A total of 24 samples were sent to Tianjin Novogene Co., Ltd. for RNA sequencing.

Transcriptome analysis

Bowtie V2.2.3 was utilized to establish a reference genome index (Langmead & Salzberg, 2012) based on the latest orchardgrass genome data published by our group (Huang et al., 2020). The double-ended clean-read sequences were compared with the reference genome using TopHat V2.0.12 (Kim et al., 2013). Then, the number of reads relative to each gene was calculated using HTSeq V0.6.1 (Anders, Pyl & Huber, 2015), and the FPKM or TPM value of each gene was calculated according to the length of the gene and the number of reads per gene. If the expression level ratio between the experimental group and the control group was greater than 1.5 or less than 1.5−1, we considered that there was a differential expression (Data with zero or infinite ratios are deleted).

GO and KEGG enrichment analysis

Gene Ontology (GO) and Kyoto Encyclopedia of Genes and Genomes (KEGG) enrichment analyses of DgWRKYs were performed using OmicShare tools (https://www.omicshare.com/tools/).

Identification of RGAs and analysis of cis-acting elements

Identification of NBS-LRRs

The IDs of the members of the NBS-LRR family were obtained from the NBS gene family articles published by our research group (Ren et al., 2020).

Identification of LRR-RLPs

A. thaliana RLP sequences were downloaded in order to BLASTP LRR-RLP family sequences. PFAM domain search (http://pfam.xfam.org/search#tabview=tab1) was used to search for a sequence structure that does not contain the protein kinase (PKinase) domain structure but does contain the LRR structure domain sequence. TMHMM-2.0 (https://services.healthtech.dtu.dk/service.php?TMHMM-2.0) was used to identify whether the sequences contain a transmembrane domain structure. SignalP-5.0 (https://services.healthtech.dtu.dk/service.php?SignalP-5.0) was used to identify the presence of a signal peptide.

Identification of LRR-RLKs

RLK sequences were downloaded from NCBI, and BLASTP was performed using them. The genes containing LRR and PKinase domains were screened out using the PFAM database, and transmembrane domains and signaling proteins were searched for using TMHMM-2.0 and SignalP-5.0, respectively.

Prediction of cis-acting elements of RGA genes

The orchardgrass genome sequence was downloaded from the orchardgrass genome database, and the 1.5-kb nucleic acid sequences upstream of each gene were extracted using TBtools. The identified sequences are shown in Table S4. Then, PlantCARE (http://bioinformatics.psb.ugent.be/webtools/plantcare/html/) was used to predict cis-acting element.

WGCNA and pearson correlation coefficient

Pearson correlation coefficients were determined and WGCNA was conducted between WRKY and RGAs expression levels based on the expression of orchardgrass under rust infection. If a correlation was greater than 0.8 or less than -0.8 and the weighted correlation coefficient was greater than 0.5, we deemed that correlation to be a very strong correlation.

Results

Identification and chromosomal locations of WRKYs in orchardgrass

A total of 93 protein sequences with a WRKY domain were identified by BLASTP and/or HMMER. These identified proteins are encoded by genes located on all seven chromosomes (Fig. 1), except for three unmapped genes, and most of them are distributed on chromosomes 5 and 6, which contain 24 and 18 genes, respectively. Chromosomes 2 and 7 contain the fewest WRKY genes, only seven each. Based on their chromosomal locations, these 90 mapped DgWRKY genes were named from DgWRKY1 to DgWRKY90, while the remaining 3 unmapped DgWRKY genes were named DgWRKY0-1, DgWRKY0-2, and DgWRKY0-3. Then, the online tool Search Pfam was used to further retrieve the conserved domains and the locations of all sequences (Table S5). This analysis revealed that in addition to a WRKY domain, DgWRKY86, 37, 39, 40, 5, and 11 also contain a Plant zinc cluster domain (PF10533), while DgWRKY84 and 6 contains a Rx N-terminal domain (PF18052), and DgWRKY6 contains a NB-ARC domain (PF00931). As DgWRKY6 contains both NB-ARC and WRKY disease-resistant domains, NB-ARC is the characteristic domain of the NBS family, we conducted an online BLASTP of DgWRKY6 and found that it is a homolog of RPM1, a disease-resistance gene from Triticum urartu, and is therefore likely to be involved in disease resistance in orchardgrass.

Figure 1 The location of the WRKY gene family on different orchardgrass chromosomes.

The chromosome number is indicated at the top of the figure.

Protein physical and chemical properties analysis and subcellular localization prediction

All of these proteins were analyzed using the ProtParam tool, which estimated amino acid length, MW, pI, the instability index, the aliphatic index, and GRAVY for each sequence. Meanwhile, WoLF PSORT was utilized to predict protein subcellular localization (Table 1). There were 51 (54.84%) protein sequences with pI estimates less than 7: 82 (88.17%) were located in the nucleus, 7 in the chloroplast, 2 in the cytoplasm, 1 in mitochondria, and 1 in the peroxisome. Only five sequences were estimated to be stable (i.e., instability index <40). The chromosome locations, WRKY domains, zinc finger motifs, and gene lengths of each protein were also determined (Table S6).

Table 1 DgWRKY proteins’ physical and chemical properties and subcellular localization prediction.

Gene name	Group	Localization	length	MW	pI	Instability index	Aliphatic index	GRAVY	
DgWRKY10	I	nucl	759	81174.33	6.09	51.60	58.05	−0.630	
DgWRKY29	I	nucl	690	74783.88	6.10	58.05	50.78	−0.742	
DgWRKY30	I	nucl	666	72750.32	6.37	52.11	66.76	−0.599	
DgWRKY3	I	nucl	610	65850.26	6.56	56.53	52.64	−0.782	
DgWRKY22	I	nucl	588	64690.50	7.31	47.31	73.49	−0.501	
DgWRKY57	I	nucl	570	61042.40	6.33	58.08	39.51	−0.842	
DgWRKY85	I	chlo	519	56871.93	8.98	50.44	61.10	−0.852	
DgWRKY33	I	nucl	505	53501.74	8.59	63.27	53.21	−0.823	
DgWRKY9	I	nucl	503	53911.84	5.81	59.34	40.66	−0.782	
DgWRKY16	I	nucl	486	51550.55	8.73	60.52	55.76	−0.741	
DgWRKY80	I	nucl	477	50836.97	8.33	48.03	54.51	−0.717	
DgWRKY2	I	nucl	419	45348.61	5.96	48.94	71.03	−0.568	
DgWRKY35	I	nucl	412	44569.95	6.97	61.23	46.53	−0.970	
DgWRKY45	IIa	nucl	866	90306.68	7.97	51.32	56.79	−0.502	
DgWRKY31	IIa	nucl	594	62057.86	8.93	44.26	60.56	−0.449	
DgWRKY12	IIa	nucl	553	58281.66	5.11	51.23	55.70	−0.755	
DgWRKY67	IIa	chlo	553	57730.58	6.65	6.65	65.79	−0.416	
DgWRKY47	IIa	nucl	526	56343.84	7.38	53.54	59.13	−0.624	
DgWRKY43	IIa	nucl	500	52272.58	7.22	52.80	66.54	−0.389	
DgWRKY13	IIa	nucl	344	37356.00	7.74	53.82	66.48	−0.668	
DgWRKY90	IIa	nucl	338	36482.79	9.13	51.49	64.47	−0.679	
DgWRKY27	IIa	nucl	321	34562.87	6.47	56.48	60.25	−0.593	
DgWRKY26	IIa	nucl	321	34562.87	6.47	56.48	60.25	−0.593	
DgWRKY28	IIa	nucl	272	29343.17	6.60	48.49	77.87	−0.342	
DgWRKY51	IIb	nucl	379	39772.32	8.54	46.83	54.72	−0.610	
DgWRKY49	IIb	nucl	346	37879.06	6.67	63.51	50.20	−0.845	
DgWRKY69	IIb	nucl	325	34783.42	6.92	57.52	51.78	−0.810	
DgWRKY36	IIb	nucl	311	32487.14	8.36	59.73	55.59	−0.463	
DgWRKY4	IIb	nucl	283	30016.45	6.04	44.73	58.80	−0.490	
DgWRKY72	IIb	nucl	268	28923.51	9.85	54.48	67.87	−0.461	
DgWRKY42	IIb	nucl	258	27837.28	9.72	49.03	48.91	−0.721	
DgWRKY48	IIb	cyto	236	25755.53	5.97	55.33	55.47	−0.644	
DgWRKY53	IIb	chlo	233	24697.45	8.71	57.54	55.79	−0.530	
DgWRKY8	IIb	nucl	232	25259.58	8.55	38.58	62.67	−0.449	
DgWRKY52	IIb	chlo	231	24452.04	7.01	63.53	53.33	−0.544	
DgWRKY19	IIb	nucl	229	25943.34	8.79	51.93	51.05	−0.641	
DgWRKY66	IIb	nucl	217	23544.33	9.58	59.34	52.44	−0.653	
DgWRKY55	IIb	nucl	214	23894.85	5.66	39.96	62.94	−0.654	
DgWRKY73	IIb	nucl	214	23315.11	6.65	42.03	71.59	−0.593	
DgWRKY44	IIb	nucl	211	22099.16	8.37	44.98	49.34	−0.505	
DgWRKY83	IIb	nucl	195	21231.25	7.05	57.37	47.18	−0.730	
DgWRKY15	IIb	nucl	189	21447.50	8.39	46.42	43.28	−1.025	
DgWRKY7	IIc	nucl	482	51349.19	6.13	49.38	48.69	−0.645	
DgWRKY0-1	IIc	nucl	465	49015.58	6.13	48.05	53.68	−0.496	
DgWRKY40	IIc	nucl	404	43627.13	9.37	60.10	62.87	−0.582	
DgWRKY37	IIc	nucl	386	41546.18	10.11	55.96	65.00	−0.578	
DgWRKY89	IIc	nucl	375	39789.76	4.91	65.70	50.29	−0.761	
DgWRKY41	IIc	nucl	362	39076.64	6.43	47.28	62.82	−0.540	
DgWRKY32	IIc	nucl	362	38951.35	10.01	55.14	60.39	−0.561	
DgWRKY11	IIc	pero	342	36213.18	9.83	49.99	67.13	−0.427	
DgWRKY79	IIc	nucl	339	37029.67	6.19	50.30	65.63	−0.569	
DgWRKY39	IIc	nucl	336	36699.61	9.79	46.81	62.44	−0.616	
DgWRKY14	IIc	nucl	332	35413.52	6.08	64.66	53.61	−0.640	
DgWRKY74	IIc	nucl	326	35117.48	6.46	52.00	60.25	−0.609	
DgWRKY5	IIc	nucl	317	33740.11	9.61	51.51	57.03	−0.636	
DgWRKY54	IIc	nucl	309	33685.98	4.83	81.08	49.94	−0.917	
DgWRKY86	IIc	nucl	299	31307.43	10.05	54.79	64.78	−0.454	
DgWRKY70	IIc	nucl	296	31660.20	5.84	55.63	63.01	−0.525	
DgWRKY71	IIc	nucl	294	31792.51	5.16	54.38	67.35	−0.509	
DgWRKY56	IIc	nucl	292	31133.43	5.13	58.90	54.11	−0.666	
DgWRKY65	IId	nucl	357	40728.89	9.98	73.05	47.90	−0.959	
DgWRKY64	IId	nucl	322	34437.13	6.22	58.32	58.29	−0.744	
DgWRKY38	IId	nucl	94	10639.82	9.33	36.45	41.49	−1.180	
DgWRKY84	III	nucl	1114	126056.21	8.49	46.14	89.43	−0.274	
DgWRKY6	III	cyto	973	111435.07	6.46	40.31	101.39	−0.167	
DgWRKY82	III	chlo	450	49425.21	9.12	52.21	63.60	−0.578	
DgWRKY81	III	nucl	388	41378.33	6.05	54.13	67.76	−0.388	
DgWRKY46	III	nucl	360	39955.21	8.29	62.40	70.72	−0.453	
DgWRKY61	III	nucl	359	39654.04	6.26	56.12	67.60	−0.369	
DgWRKY1	III	nucl	353	37047.26	8.92	62.13	53.43	−0.442	
DgWRKY87	III	chlo	350	38159.62	7.48	62.28	64.17	−0.498	
DgWRKY50	III	nucl	339	36124.25	6.70	48.39	61.15	−0.442	
DgWRKY24	III	nucl	338	36343.56	5.39	55.23	63.02	−0.405	
DgWRKY20	III	nucl	337	35423.42	6.60	51.34	63.23	−0.379	
DgWRKY21	III	nucl	331	35153.98	6.24	53.49	61.36	−0.447	
DgWRKY25	III	nucl	330	35600.91	6.66	57.29	62.67	−0.436	
DgWRKY60	III	nucl	315	33705.31	5.34	54.52	66.10	−0.459	
DgWRKY0-2	III	nucl	314	33810.94	6.09	54.55	72.42	−0.361	
DgWRKY0-3	III	nucl	314	33810.94	6.09	54.55	72.42	−0.361	
DgWRKY17	III	nucl	311	32769.88	5.79	56.51	57.46	−0.539	
DgWRKY23	III	nucl	306	33359.39	6.06	54.35	67.61	−0.430	
DgWRKY75	III	nucl	303	32361.68	5.25	49.32	65.54	−0.457	
DgWRKY77	III	nucl	300	32191.45	5.10	52.75	62.30	−0.468	
DgWRKY58	III	nucl	299	31712.53	5.31	54.30	67.66	−0.305	
DgWRKY78	III	nucl	293	32354.14	6.52	70.16	61.71	−0.668	
DgWRKY59	III	nucl	272	28658.00	7.00	55.22	64.38	−0.324	
DgWRKY88	III	chlo	268	29925.58	8.70	49.93	64.48	−0.651	
DgWRKY62	III	nucl	267	29921.99	5.84	61.26	44.27	−0.870	
DgWRKY63	III	nucl	261	28685.06	6.54	48.53	63.22	−0.546	
DgWRKY68	III	mito	246	26701.98	6.59	38.64	67.15	−0.478	
DgWRKY76	III	nucl	238	26066.07	8.78	74.88	63.70	−0.720	
DgWRKY34	III	nucl	220	24507.97	7.61	66.38	43.55	−0.838	
DgWRKY18	III	nucl	206	22874.32	8.44	58.24	46.99	−0.862	
Note:

MW, pI, GRAVY, Cyto, Nucl, Chlo, pero, and mito represent molecular weight, theoretical isoelectric points, grand average of hydropathicity cytoplasm, nucleus, chloroplast, peroxisome, and mitochondria, respectively.

Phylogenetic analysis and multiple sequence alignment of WRKY genes

A total of 335 WRKY genes from A. thaliana (71 AtWRKYs), T. aestivum (171 TaWRKYs), and D. glomerata (93 DgWRKYs) were used to construct a phylogenetic tree (Fig. 2). Based on the number of WRKY domains and the zinc finger motif that the sequences contain, these WRKYs were classified into three groups (groups I–III) (Eulgem et al., 2000). Group I contained 13 members with two WRKY domains each located on both the N-terminus and C-terminus and two zinc finger motifs of the C2H2 (CX4–5CX22–23HX1H) type (Figs. 3A–3B). Among them, one conserved domain of DgWRKY30 was mutated from WRKYGQK to WRKYGKR (Fig. 3A). Furthermore, according to the tree obtained, group II was classified into four subgroups, group IIa, group IIb, group IIc, and group IId (Fig. 2), which possessed 11, 18, 18, and 3 members, respectively. Each member of group II had one WRKY domain and one zinc finger motif (Figs. 3C–3F) of the C2H2 type. In group IIb, DgWRKY55 carried an incomplete WRKY conserved sequence, and DgWRKY44, 48, 52, 53, 72, and 83 exhibit a WRKYGKK variant sequence. Except for DgWRKY89, the others in groups IIa and IIc contain the CX5CX23HX1H motif while proteins in groups IIb and IId contain CX4CX23HX1H zinc finger motifs, except for DgWRKY38 and DgWRKY52. DgWRKY38 and DgWRKY52 had CX4CX22HX1H and CX4GX23HX1H zinc finger motifs, respectively (Fig. 2). Thirty sequences belong to group III, each with one WRKY domain and one zinc finger motif (Figs. 3G–3H) of type C2HC (CX7CX23-28HX1C); however, DgWRKY6 had a different zinc finger motif sequence, CX7CX23HX1Y. Additionally, DgWRKY87 was identified to carry a WRKY domain, but we manually retrieved it and found that it lacked a WRKYGQK heptapeptide (Fig. 3G); accordingly, we defined it as a WRKY-like gene.

Figure 2 Phylogenetic tree obtained for the WRKY TF family members in, orchardgrass, wheat and Arabidopsis.

Different colors represent different sub-classes in the WRKY gene family.

Figure 3 Multiple sequence alignments of the WRKY domains of from DgWRKYs.

The group name is indicated at the left of the figure.—N represents the N-terminal WRKY domains,—C represents the C-terminal WRKY domains. (A) Multiple sequence alignments of the group I N-terminal. (B) Multiple sequence alignments of the group IⅠ C-terminal. (C) Multiple sequence alignments of the group IIa. (D) Multiple sequence alignments of the group IIb. (E) Multiple sequence alignments of the group IIc. (F) Multiple sequence alignments of the group IId. (G) Multiple sequence alignments of the group III N-terminal. (H) Multiple sequence alignments of the group III C-terminal.

Gene structures and consensus motifs of WRKYs in orchardgrass

A phylogenetic tree containing DgWRKYs using the maximum likelihood method was constructed using MEGA X (Fig. 4A). For this, we analyzed the consensus motifs determined by MEME and TBtools. By setting retrieval parameters, the distributions of 10 types of motifs in DgWRKYs were determined (Fig. 4B), and the members from the same subgroup had similar conserved motifs. All proteins contained motif 1, with a conserved WRKY amino acid sequence, except for DgWRKY87 (motif logos shown in Fig. S1). Meanwhile, apart from some members of group III (DgWRKY21, 20, 87, 88, 61, 46, 60, 62, 63, 34, 18, 78, 76), all others contained motif 2, which shares sequence identity with a zinc finger motif. Motif 3, representing the C2H2 zinc finger structure, is distributed across all groups but group III. Motif 4 is mainly distributed in groups I and IIb, while motif 5 is mainly distributed in group I but also in some parts of group IIc. Motif 6 is mainly distributed in groups IIa and III. Motifs 7, 8, 9, and 10 are mainly distributed in groups IIa, III, IIc, and III, respectively. By combining the analysis of motifs, it was determined that motifs 1, 2, and 3 represent part of the C2H2 structure while motifs 1, 2, and 10 constituted part of the C2HC structure.

Figure 4 The phylogenetic tree, conserved motifs, and gene structure of orchardgrass WRKY family.

(A) Phylogenetic tree of WRKY proteins constructed by MEGA using the ML (Maximum likelihood) method. (B) The motifs of WRKY protein are displayed in the figure. Different motifs are denoted by different colors numbered from motif 1–10 at the top right panel of the figure. The detailed information of the 10 motifs is listed in Fig. S1. (C) The gene structure of 93 orchardgrass WRKY genes. The green boxes, yellow box, and full line represent CDS (Sequence coding for amino acids in protein), UTR (Untranslated region), and introns respectively.

For further identification of the phylogenetic relationships among DgWRKYs, the position of the CDS and untranslated region (UTR) for each protein was determined (Fig. 4C). There were nine genes (9.68%) that lacked an intron (DgWRKY83, 44, 13, 68, 0–2, 0–3, 77, 75, 86), six of which (DgWRKY44, 68, 0–2, 0–3, 77, 75) had no UTR. Additionally, 14 genes (15.05%) had two CDSs, 46 genes (49.46%) had three CDSs, 16 genes (17.20%) had four CDSs, 5 genes (5.37%) had five CDSs, 3 genes (3.23%) had six CDSs. Overall, DgWRKYs in the same subgroup had similar genetic structures throughout orchardgrass.

Gene duplication and calculation of Ka/Ks

Gene duplication is deemed to be one of the crucial drivers of the evolution of genomes and genetic systems. Segmental and tandem duplications are considered to be the two main phenomena underlying the expansion of plant gene families (Cannon et al., 2004). To study the duplication of WRKY genes throughout the evolution of orchardgrass, BLASTN and McscanX were used to perform a comparison of homologs, and 42 genes (45.16%) were determined to be involved in duplication events, including 34 segmental duplicate genes (Fig. 5A) and 14 tandem duplicate genes (Some genes are both tandem duplicates and segmental duplicates). Six tandem duplicate gene pairs (DgWRKY20 & DgWRKY21, DgWRKY26 & DgWRKY27, DgWRKY52 & DgWRKY53, DgWRKY58 & DgWRKY59, DgWRKY75 & DgWRKY77, DgWRKY76 & DgWRKY78) are shown in Fig. 5B to be distributed on chromosomes 3, 5, and 6. However, DgWRKY0-2 & DgWRKY0-3 also comprise a pair of duplicate genes, though they cannot be localized to any chromosome in the published genome. The pairs DgWRKY26 & DgWRKY27 and DgWRKY0-2 & DGWRKY0-3 were identical duplicates, and each member of a pair shared the same amino acid sequence. The Ka/Ks values of the four pairs of tandem duplicate genes were calculated (Table 2), and their values were low (i.e., Ka/Ks < 0.5), which indicates that these genes have been subjected to purifying selection (Ziheng & Rasmus, 2000).

Figure 5 Genomic locations of tandem and segmentally duplicated gene pairs in the orchardgrass genome.

(A) Gray lines in the background indicate the synteny blocks within the whole orchardgrass genome, and red lines denote the segmental duplication of WRKY gene pairs. (B) Red lines denote the tandem duplication WRKY gene pairs, the gene name has been labeled.

Table 2 The Ka/Ks of 4 tandem repeat gene pairs.

Gene name	Ka	Ks	Ka/Ks	Length	
DgWRKY52 & DgWRKY53	0.050	0.149	0.337	687	
DgWRKY76 & DgWRKY78	0.032	0.091	0.352	705	
DgWRKY75 & DgWRKY77	0.052	0.149	0.346	894	
DgWRKY20 & DgWRKY21	0.084	0.255	0.330	984	

Expression profile analysis of WRKY genes across orchardgrass tissues

It has been reported that WRKY genes are expressed in a variety of cell types and under different physiological conditions, enabling it to participate in the regulation of a variety of biological processes (Eulgem et al., 2000). In order to elucidate the expression pattern of WRKYs in different tissues of orchardgrass, a total of 72 expression profiles of WRKY genes in root, stem, leaf, flower, and spike tissues were obtained (Fig. 6). Most genes (44) were found to have the highest expression level in root samples, followed by spike samples (11 genes), while the leaf had the fewest genes with the highest observed expression level (4). Thus, it was obvious that WRKY genes are preferentially expressed in roots over leaves (Fig. 7). Notably, Ji et al. (2014) found that persistent drought damaged the leaves more than the roots in orchardgrass.

Figure 6 Expression of DgWRKY genes in root, stem, leaf, spike, flower tissues.

Blue to red color denotes low to high relative expression. The original expression values were normalized by Z-score normalization.

Figure 7 The number of the gene with the highest expression level in root, stem, leaf, spike, flower tissues.

Expression profile analysis of WRKY genes in orchardgrass under different abiotic stresses

Members of the WRKY TF family are widely involved in the regulation of abiotic stresses in plants (Jiang et al., 2017). To further explore the potential functions of DgWRKY genes under various abiotic stresses, the expression patterns of DgWRKY genes under heat, drought, and submergence stress were determined. Thus, 60, 88, and 79 genes were found to be expressed under heat, drought, and submergence stress, respectively. After 10 days of heat stress, 19 genes were up-regulated. (Unless otherwise specified, the threshold for up-regulation and down-regulation in this study is 1.5-fold change, where the fold change is the ratio of FPKM or reads per kilobase per million mapped reads (RPKM) values). Additionally, 13 genes were down-regulated under heat treatment compared to the control in the heat-resistant ‘BAOXING’ cultivar (Fig. 8A). Additionally, three genes were discovered to be up-regulated by a more than 5-fold change, while DgWRKY73 was the most up-regulated (20-fold). At the same time, 31 genes were down-regulated, and only one gene (DgWRKY41) was up-regulated in the heat-susceptible ‘01998’ cultivar. After 26 days of heat stress, 9 and 16 genes were up-regulated and down-regulated, respectively, in ‘BAOXING,’ while 5 and 26 genes were up-regulated and down-regulated, respectively, in ‘01998.’ DgWRKY20 was up-regulated at 10 days and 26 days in ‘BAOXING,’ but down-regulated in ‘01998.’ Under heat stress, 22 genes only differentially expressed in ‘BAOXING’ after 10 days (Fig. 9A) and 12 genes only differentially expressed in ‘BAOXING’ after 26 days, while the express levels of none of these genes were changed in ‘01998’ at the corresponding time (Fig. 9B).

Figure 8 The expression profiles of WRKY genes (The sample names are shown at the bottom) in different abiotic stress of orchardgrass.

(A) DgWRKY expression patterns in ‘BAOXING’ (heat-resistant cultivar) and ‘01998’ (heat-susceptible cultivar) under heat stress. (B) DgWRKY expression patterns in root and leaf under drought stress in ‘BAOXING.’ (C) DgWRKY expression patterns in ‘DIANBEI’ (submergence-tolerant cultivar) and ‘ANBA’ (submergence-susceptible cultivar) under submergence stress. Blue to red color denotes low to high relative expression. The original expression values were normalized by Z-score normalization.

Figure 9 Venn diagram of different comparison groups.

(A) The comparison group of upregulated and downregulated genes in ‘BAOXING’ (heat-resistant cultivar) and ‘01998’ (heat-susceptible cultivar) at 10th day under heat stress. (B) The comparison group of upregulated and downregulated genes in ‘BAOXING’ and ‘01998’ at 26th day under heat stress. (C) The comparison group of upregulated and downregulated genes in root and leaf of orchardgrass at 18th day under drought stress. (D) The comparison group of upregulated and downregulated genes in ‘DIANBEI’ (submergence-tolerant cultivar) and ‘ANBA’ (submergence-susceptible cultivar) at 8th hour under submergence stress. (E) The comparison group of upregulated and downregulated genes in ‘DIANBEI’ and ‘ANBA’ at 24th hour under submergence stress. (F) The comparison group of upregulated and downregulated genes in PI292589 (rust-susceptible line) and PI251814 (rust-resistant line) at 7th day under rust stress. (G) The comparison group of upregulated and downregulated genes in PI292589 and PI251814 at 14th day under rust stress. (H) The comparison group of DEGs (differentially expressed genes) under heat, drought, submergence and rust stress.

The expression profiles of WRKY genes in leaf and root tissues from ‘BAOXING’ under drought stress were also assessed (Fig. 8B). After 18 days of drought treatment, 7 and 41 genes were up-regulated and downregulated, respectively, in leaf tissue, while 23 and 31 genes were up-regulated and down-regulated, respectively in root tissue. DgWRKY64 was upregulated, with 2-fold and 4-fold changes observed in leaf and root tissues, respectively. The expression levels of DgWRKY18, DgWRKY49, DgWRKY51, and DgWRKY64 were increased under both heat stress and drought stress. Additionally, 7 and 13 genes were up-regulated and down-regulated, respectively, in root tissue only, but had no significantly changed in leaves (Fig. 9C).

The expression profiles of orchardgrass at 8, and 24 h of submergence were constructed (Fig. 8C). Compared with the control group, 41 and 8 genes were up-regulated and down-regulated, respectively, in the submergence-tolerant ‘DIANBEI’ cultivar at 8 h after submergence stress; among these, 12 up-regulated genes and 4 down-regulated genes were only differentially expressed in ‘DIANBEI’ (Fig. 9D). By 24 h of the submergence stress treatment, 46 and 9 genes were up-regulated and down-regulated, respectively, in ‘DIANBEI,’ and 15 up- and 3 down-regulated genes were only differentially expressed in ‘DIANBEI’ (Fig. 9E). For the submergence-susceptible ‘ANBA’ cultivar, 40 and 6 genes were up-regulated and down-regulated after 8 h of submergence stress, while 39 and 10 genes were up-regulated and down-regulated after 24 h of submergence stress.

Expression profile analysis of WRKY genes in orchardgrass under biotic stress

WRKY TFs are also often involved in biotic stress responses (Jiang et al., 2017). To understand the WRKY genes involved in plant responses to biotic stress, we measured the expression levels of orchardgrass under rust infection and found 73 genes were related to rust stress (Fig. 10A). After 7 days of rust infection, 51 and 3 genes were up-regulated and down-regulated, respectively, in the highly rust-susceptible PI292589 line, while 53 and 5 genes were up-regulated and down-regulated in the highly rust-resistant PI251814 line. Additionally, 2 and 14 genes down-regulated and up-regulated, respectively, in PI251814 only (Fig. 9F). After 14 days of rust infection, 2 and 52 genes were up-regulated and down-regulated, respectively, in PI292589, while 47 genes and zero genes were up-regulated and down-regulated, respectively, in PI251814. Six of up-regulated genes exhibited varied expression levels only in PI251814 (Fig. 9G). The expression of most WRKY genes in susceptible and resistant plants were up-regulated on the 7th day of rust infection. However, as stress duration increased, only two genes were up-regulated in susceptible plants by the 14th day, while 47 genes were up-regulated in resistant plants. In order to prove the reliability of the data, some significance analyses were conducted on WRKY expression. As shown in Fig. 10B, there was a significant difference between PI251814 and PI292589. Significant difference existed between PI251814 before and after inoculation, but non-significant difference existed between PI292589 before and after inoculation (Fig 10C). There was no significant difference between high rust-resistant line and high rust-susceptible line before inoculation, but there was a significant difference after inoculation (Fig 10C). These results indicated that WRKY expression levels between PI251814 and PI292589 had little difference before inoculation with rust, but great changes occurred after inoculation with rust.

Figure 10 The expression profiles of WRKY genes in rust stress of orchardgrass, and box plot for significance test.

(A) The expression profiles of WRKY genes in rust stress of orchardgrass. The sample names are shown at the bottom. Blue to red color denotes low to high relative expression. The original expression values were normalized by Z-score normalization. (B) The box plot for significance test between PI292589 and PI251814. The more * symbols, the more significant the difference. (C) The box plot for significance test between HR_CK, HR_R, HS_CK, HS_R. HR stands for the highly rust-resistant PI251814 line. HS stands for the highly rust-susceptible PI292589 line. CK stands for no rust inoculation. R stands for rust inoculated. The more * symbols, the more significant the difference.

Notably, there were 53, 68, 70, and 73 differentially expressed genes (DEGs) at a 1.5-fold change threshold in the heat, drought, submergence, and rust stress treatments, respectively (Fig. 11). Relative to the control treatment, more DEGs were observed in the rust stress treatments than under other treatments. There were 37 common DEGs under all stresses, and 4 genes were differentially expressed only under rust stress (Fig. 9H).

Figure 11 The number of DEGs (Differentially expressed genes) under heat, drought, submergence and rust stress.

GO and KEGG enrichment analyses

To further elucidate the biological function and molecular mechanisms of WRKY TFs, GO enrichment analysis and KEGG enrichment analysis of 93 genes were conducted. As seen in Fig. 12, the most genes were enriched for the terms membrane and membrane part among cell components; biological regulation, cellular process, metabolic process, and regulation among biological processes; and binding and nucleic acid binding TF activity among molecular functions. Unexpectedly, through KEGG enrichment analysis, we found that all the genes clustered into just three pathways: plant–pathogen interaction, mitogen-activated protein kinase (MAPK) signaling pathway-plant, and aminoacyl-tRNA biosynthesis (Fig. 13). It is well known that the first two pathways are generally associated with plant responses to environmental stress. Most were involved in plant–pathogen interaction, corresponding to 78 genes (83.87%). Among them, 59 genes were differentially expressed under rust stress. This suggests that most WRKY TFs in orchardgrass are related to responses to pathogen interactions.

Figure 12 GO (Gene Ontology) categories of the target genes for 93 DgWRKYs.

Figure 13 KEGG (Kyoto encyclopedia of genes and genomes) categories of the target genes for 93 DgWRKYs.

Identification of RGAs and their cis-acting elements

According to the above results, DgWRKYs may be related to biotic stress responses, but how they participate in the defense process has been unclear. As we know, R genes are usually involved in plant defense against pathogens (McHale et al., 2006). Therefore, investigating whether DgWRKYs can interact with RGAs is of substantial importance. Using BLASTN and HMM SEARCH, 281 RGAs were identified, including 65 NBS-LRRs, 169 LRR-RLKs, and 47 LRR-RLPs. The NBS-LRR gene family was identified by Ren et al. (2020) in orchardgrass. W-box elements are the specific binding site of WRKYs (Eulgem et al., 2000). To determine whether WRKY TFs can participate in the response to rust stress by regulating the expression of RGAs, 1.5-kb nucleic acid sequences upstream of RGAs were extracted for prediction of cis-acting elements. As shown in Fig. 14, 154 of the 281 RGAs have W-box elements, with DG3C03551.1 containing the most W-box elements (4).

Figure 14 Predicted cis-elements in RGAs promoters.

Promoter sequences (−1,500 bp) of 218 RGAs (just 154 have W-box) are analyzed by PlantCARE.

WGCNA and co-expression analyses of WRKYs with RGAs

WGCNA and correlation analyses were also performed for WRKY and RGAs. As shown in Fig. 15, principal component analyses (PCAs) showed that the high-resistance plants 7 and 14 days after rust inoculation (HR_7 and HR_14, respectively) were quite distinct from other plants. This illustrates the difference between high-resistance and high-sensitivity plants. A total of nine modules were identified by WGCNA. Among the modules shown in Fig. 16, the turquoise and brown ones were significantly correlated with HR_7 and HR_14, respectively. Therefore, we extracted the data associated with these two modules for further analysis, finding 1261 pairs of interactions, among which we removed low reliability interactions (weight < 0.5), thus retaining the high reliability WRKY-RGA interactions. Additionally, the RGAs without a W-box element were removed. Pearson correlation coefficients describing the relationship between the expression levels of each gene pair were calculated for verification, and a high-quality interaction map was finally obtained (Table 3). Ultimately, there were 24 interactions identified between 14 WRKYs and 5 RGAs (Fig. 17). It is worth noting that DG6C02319.1 (LRR-RLK family member) was determined to potentially interact with all 14 WRKYs, for which we proposed several hypotheses. One possibility is that DG6C02319.1 can induce the expression of WRKYs after inoculation with rust fungus to resist rust fungus invasion. The second is that DG6C02319.1 is not the only gene that can induce the expression of WRKYs, as WRKYs still regulate the transcription of DG6C02319.1 (because the promoter region of DG6C02319.1 contains a W-box element). The third possibility is that various WRKYs coordinate and regulate the transcription of DG6C02319.1 to participate in the immune process of plants against rust.

Figure 15 Principal component analysis induced by rust.

Figure 16 Heat map of correlation between modules and traits.

Red indicates a positive correlation, blue indicates a negative correlation, and the weighted correlation coefficient and P value are indicated in the box.

Figure 17 Interactions between 14 WRKYs and 5 RGAs.

Table 3 Pearson correlation coefficient for pairs.

DgWRKYs	RGAs	cor	P-value	
DgWRKY11.1	DG6C02319.1	0.965717222	5.48582E−12	
DgWRKY13.1	DG6C01662.1	0.966808979	4.11599E−12	
DgWRKY13.1	DG6C02319.1	0.972198837	8.51905E−13	
DgWRKY14.1	DG6C02319.1	0.962503072	1.21456E−11	
DgWRKY23.1	DG6C02319.1	0.841006826	3.41018E−06	
DgWRKY24.1	DG6C01662.1	0.930842849	2.6691E−09	
DgWRKY24.1	DG6C02319.1	0.961464602	1.54719E−11	
DgWRKY24.1	DG3C03908.1	0.964729314	7.05855E−12	
DgWRKY36.1	DG6C02319.1	0.935563084	1.43759E−09	
DgWRKY50.1	DG2C02110.1	0.895697806	9.45072E−08	
DgWRKY50.1	DG6C02319.1	0.956800712	4.25282E−11	
DgWRKY50.1	DG3C03908.1	0.96377025	8.95481E−12	
DgWRKY50.1	DG3C00088.1	0.960773767	1.81091E−11	
DgWRKY57.1	DG2C02110.1	0.901223329	5.91198E−08	
DgWRKY57.1	DG6C02319.1	0.953225134	8.58626E−11	
DgWRKY57.1	DG3C03908.1	0.966609051	4.34141E−12	
DgWRKY57.1	DG3C00088.1	0.949987128	1.54974E−10	
DgWRKY59.1	DG6C02319.1	0.823333144	8.22376E−06	
DgWRKY60.1	DG6C02319.1	0.901864922	5.58866E−08	
DgWRKY74.1	DG6C02319.1	0.866684293	7.70972E−07	
DgWRKY81.1	DG6C02319.1	0.914834278	1.63815E−08	
DgWRKY9.1	DG6C02319.1	0.939321965	8.48671E−10	
DgWRKY9.1	DG3C03908.1	0.955631067	5.38563E−11	
DgWRKY90.1	DG6C02319.1	0.871137929	5.77559E−07	

Discussion

Animal husbandry depends on the feed industry, but feed resources have become a major constraint on animal production in Asia (Devendra & Sevilla, 2002). Orchardgrass is one of the top four perennials forage crops globally. It plays an important role in the production of meat and dairy products (Wilkins & Humphreys, 2003), but its yield is threatened by various environmental stresses. A large number of studies have shown that WRKY TFs can regulate a variety of plant processes and responses to various biotic and abiotic stresses (Cai et al., 2021; Wang et al., 2020b; Wang et al., 2020c; Xiang et al., 2021). In this study, WRKY family members were first identified from orchardgrass, and all sequences were located, classified, and analyzed according to their expression profiles, which is of great significance for molecular-assisted breeding of orchardgrass.

A total of 93 DgWRKY coding genes were identified and divided into three groups according to the number of WRKY domains and the type of zinc finger structure carried in the protein sequence. Group II had the most members, which was consistent with previous reports of WRKYs in A. thaliana, M. esculenta, G. max, and Caragana intermedia (Bencke-Malato et al., 2014; Eulgem et al., 2000; Wan et al., 2018; Wei et al., 2016). The WRKYGQK heptapeptide stretch is considered to be an important sequence for identifying and binding the W-box element of the target gene promoter (Rushton et al., 2010). In orchardgrass, some TFs exhibit variants differing from the WRKYGQK heptapeptide stretch, including WRKYGEK, WRKYGKK, and WKKYGQK (Fig. 3), and certain TFs (i.e., DgWRKY55, WRKSYYR; Fig. 3) did not contain the full WRKY amino acid sequence or even had no WRKY (i.e., DgWRKY87, Fig. 3) tetrapeptide. As far as we know, WRKSYYR is a WRKY gene family variant unique to orchardgrass. Additionally, all six TFs (DgWRKY44, 48, 52, 53, 72, 83) exhibiting the WRKYGKK heptapeptide variant belong to group IIb (Fig. 3). Research has shown that WRKY TFs with a variant heptapeptide may recognize binding elements outside W-box elements; this includes WRKYGKK, which can specifically bind with WK-box elements (TTTTCCAC) (Van Verk et al., 2008; Zhou et al., 2008). Four TFs (DgWRKY18, 34, 76, 78) with the WRKYGEK heptapeptide variant belong to group III and are located in the nucleus, and they were upregulated during drought and/or submergence stresses. Moreover, the expression of DgWRKY18 increased after treatments with each of the three abiotic stresses. The establishment of these variant sequences in WRKY gene family members correspond with long-term gene family evolution. WKKYGQK variants were distributed in groups IId and III. At the same time, we also found that most duplicate WRKY genes in orchardgrass (90.48%) belonged to groups III and II, and it is obvious from the tree in Fig. 2 that members of group III are divided into two branches, which indicates that the WRKY genes of these two branches possibly originated from different ancestors. Overall, groups II and III were more diverse, consistent with the results of Zhang & Wang (2005).

Studies have shown that the expression of genes with short introns or short total intron length are increased in plants (Chung et al., 2006). Among all WRKY genes identified in orchardgrass, nine genes had no introns. The expression levels of five of them in roots, stems, leaves, flowers, and spikes were measured. We found that three of these five genes (DgWRKY13, 44, 83) had higher FPKM values (Fig. 6).

A recent study by Mohanta, Park & Bae, 2016 has suggested that WRKY TFs may also contain more than two WRKY domains or other domains, including ZF SBP, LRR, and PAH domains. We also found that some DgWRKY TFs possess some other conserved domains, for example, Plant zinc cluster, Rx N-terminal, NB-ARC domains (Table S5).

As is widely known, TFs are usually involved in transcriptional regulation in the nucleus (Brownawell et al., 2001), which is consistent with our results. Moreover, 88.7% of DgWRKYs are located in the nucleus (Table 1), and GO enrichment analysis has also showed that most TFs were enriched in the membrane, nucleic acid binding TF activity, and biological regulation categories (Fig. 12). Based on GO enrichment analysis, some WRKY TFs are located in chloroplasts, the cytoplasm, mitochondria, and peroxisomes, while some even may be involved in transcriptional regulation outside of the cytoplasm (Table 1).

DgWRKYs were induced by various abiotic stresses, including heat, drought, and waterlogging, and they are also expressed in roots, stems, leaves, flowers, and spikes of orchardgrass, with roots and leaves exhibiting the highest and lowest expression levels respectively (Fig. 7). Additionally, the number of DEGs in root tissue was more than that in leaf tissue (Fig. 8B). Ji et al. (2014) reported that under the same intensity of drought stress, the damage to leaves was much greater than that to roots. This result indicates that DgWRKY genes play an important role in the response to drought stress and root growth.

Rust stress also changed the expression of DgWRKY genes. By comparing the number of DEGs under different stresses, we found that, relative to control conditions, the most DEGs were revealed under rust stress (Fig. 11). Through KEGG enrichment analysis of 93 WRKY transcription factors, it was found that 78 transcription factors clustered in the plant–pathogen interaction pathway (Fig. 13), among which, 59 genes were differentially expressed under rust stress. These results indicate that DgWRKYs play an important role in plant responses to rust stress. Plant responses to pathogen attacks require large-scale transcriptional reprogramming, including transcriptional reprogramming by WRKY gene family members. Many WRKY genes can negatively regulate plant defense signaling, including AtWRKY7, 11, and 17, and mutations in these genes can induce susceptibility to virulent Pseudomonas syringae (Journot-Catalino et al., 2006; Pandey & Somssich, 2009). Positive regulation of plant disease resistance signaling WRKY genes has also been observed in CaWRKY27, TaWRKY70, and WRKY22, among other genes (Cheng & Wang, 2014; Jiang et al., 2017; Wang et al., 2017). Furthermore, overexpression of WRKY22 increased resistance to Pyricularia oryzae Cav. in rice. In orchardgrass, rust-resistant plants clearly showed the most WRKY genes that increased 7 and 14 days after rust inoculation (53 and 5 genes up-regulated and down-regulated, respectively at day 7 and 47 genes and zero genes up-regulated and down-regulated, respectively, at day 14). However, in susceptible plants, after 7 days of rust stress, there were 51 genes up-regulated and just 3 genes downregulated. As the stress duration increased to 14 days, only two WRKY genes were up-regulated, while the expression of 52 genes was reduced (Fig. 10A). This suggests that most DgWRKY genes were more likely to act as positive regulators of plant disease resistance signals in orchardgrass. DgWRKY9, 13, 14, 50, and 57 showed a tendency of first being up-regulated and then down-regulated in susceptible plants, but showed a tendency of consistent up-regulation in highly resistant plants. Accordingly, these genes could be further studied as candidate genes.

DG6C02319.1 is an LRR-RLK gene, and all relevant results indicated that it is strongly co-expressed with 14 WRKY genes. It has been shown that LRR-RLK can act as a pattern recognition receptor (PRR) to stimulate PAMP-triggered immunity (PTI) in a plant by recognizing the conserved PAMP structure (Boller & Felix, 2009; Dodds & Rathjen, 2010). FLS2 is a typical LRR-RLK gene that has been confirmed encode a protein that regulates the complete MAPK signaling pathway, e.g., WRKY22/WRKY29, MKK4/MKK5, MPK3/MPK6, etc. (Thomma, Nürnberger & Joosten, 2011).

On A. thaliana chromosome 4, the promoters of many genes are rich in W-box elements, four of which encode RLKs. Both salicylic acid (SA) treatment and Pseudomonas syringae infection resulted in increased expression levels of these genes. A gel retardation assay showed that the W-box of the RLK4 promoter can be recognized by purified AtWRKY18 and SA-induced Arabidopsis nuclear extract. A further transgenic analysis shows that these W-box elements play important roles in inducing the expression of reporter genes (Du & Chen, 2000). RLK senses external stimuli and phosphorylates specific target proteins through its kinase activity, thereby transducing signals into cells (Czernic et al., 1999). These results suggest that WRKY may regulate genes encoding signal transduction proteins. The promoter regions of PR-1 and NPR1 contain W-boxes. The binding of WRKY transcription factors to PR-1 can quickly activate the signaling of early defense responses in plants, and the binding of WRKY transcription factors to NPR1 can in turn regulate NPR1 to coordinate the expression of R genes (Yu, Chen & Chen, 2001. These results are consistent with our experimental results.

Conclusions

In total, 93 WRKY genes were identified from the orchardgrass genome, and the structure of DgWRKY genes was thus revealed. DgWRKY87 is a WRKY-like gene with a structure similar to that of WRKY genes, but without the WRKYGQK heptapeptide. The physical and chemical properties and subcellular location of the identified proteins were predicted. Through the analysis of the expression profile of DgWRKYs, it was found that most DgWRKYs showed differential expression under various stresses, including heat, drought, submergence, and rust stress. This indicates that DgWRKY genes are involved in a variety of environmental stresses and processes, including biotic and abiotic stresses. Relative to control conditions, 73 DEGs (accounting for 80% of all DgWRKY genes) were observed under rust stress, and through GO and KEGG annotations, 78 WRKY TFs were observed to be enriched in pathogen interaction pathways, suggesting that WRKY genes in orchardgrass play an important role in the antibacterial defense system. Through cis-acting element prediction, WGCNA, and co-expression analysis, five RGAs and fourteen WRKYs with interactions were found. This work provides a firm foundation for further functional studies of WRKY TFs in plants.

Supplemental Information

Supplemental Information 1 The sequences for BLASTP.

Click here for additional data file.

Supplemental Information 2 Protein sequences of DgWRKYs..

Click here for additional data file.

Supplemental Information 3 Transcriptome data of WRKY in orchardgrass under different tissues and stresses, including flower, leaf, root, spike, stem, heat, drought, submergence and rust stress.

Click here for additional data file.

Supplemental Information 4 The promoter sequence of the RGAs.

Click here for additional data file.

Supplemental Information 5 The conserved domain and location of all DgWRKY sequences.

Click here for additional data file.

Supplemental Information 6 The chromosome locations, WRKY domains, zinc finger motifs and gene lengths of all DgWRKY sequences.

Click here for additional data file.

Supplemental Information 7 The logos of conserved motifs of WRKY proteins in orchardgrass.

Click here for additional data file.

Additional Information and Declarations

Competing Interests

Author Contributions

Data Availability

The authors declare that they have no competing interests.

Juncai Ren conceived and designed the experiments, performed the experiments, analyzed the data, prepared figures and/or tables, authored or reviewed drafts of the paper, and approved the final draft.

Jialing Hu analyzed the data, prepared figures and/or tables, authored or reviewed drafts of the paper, and approved the final draft.

Ailing Zhang performed the experiments, analyzed the data, authored or reviewed drafts of the paper, and approved the final draft.

Shuping Ren performed the experiments, authored or reviewed drafts of the paper, and approved the final draft.

Tingting Jing performed the experiments, authored or reviewed drafts of the paper, and approved the final draft.

Xiaoshan Wang analyzed the data, authored or reviewed drafts of the paper, and approved the final draft.

Min Sun analyzed the data, authored or reviewed drafts of the paper, and approved the final draft.

Linkai Huang conceived and designed the experiments, authored or reviewed drafts of the paper, and approved the final draft.

Bing Zeng conceived and designed the experiments, authored or reviewed drafts of the paper, and approved the final draft.

The following information was supplied regarding data availability:

The orchardgrass genome is available at Genbank BioProject: PRJNA471014.

The Transcriptome data are available at NCBI SRA: SRP158919, SRP131899, SRP049315, PRJNA565626 and PRJNA554779.

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
