# Peer review of "The whole-genome and expression profile analysis of WRKY and RGAs in Dactylis glomerata showed that DG6C02319.1 and DgWRKYs may cooperate in the immunity against rust"

_PeerJ, doi:10.7717/peerj.11919_

## Round 0.1 · original submission · Major Revisions

I agree with the reviewers in that the manuscript requires major editing. Please review the suggestions by the reviewers and make changes that were suggested. This will help to improve the manuscript significantly.

Reviewer 1 ·

Basic reporting

WRKY transcription factors (TFs) are one of the largest gene families in plants, which can regulate a variety of plant processes and widely participate in the response of plants to biotic and abiotic stresses. Here the authors identified a total of 93 DgWRKY genes and 281 RGAs, including 65, 169, and 47 NBS-LRR, LRR-RLK, and LRR-RLP. Through analyzing the expression of DgWRKY genes in orchardgrass under different environmental stress DgWRKYs responded to both biotic stress and abiotic stress. DgWRKYs and RGAs may synergistically respond to the defense process of orchardgrass against rust. This study was useful for the study of the molecular mechanisms of WRKY gene in orchardgrass.

Experimental design

Many errors occurred in the text, it should be revised.
Line 44 References should be in order of years of publication
Line 67,76,98, 106, 167, 170, 171, 172, 189, 195, 214, 274, 361, There is a problem with the citing the reference formatting.
Line 206 two parens.
Line 60, (C T 2005) and Line 633, should be Thomas Eulgen 2005
Line 420, (C & C 2002) should be Devendra C, Sevilla CC. Availability and use of feed resources in crop-animal systems in Asia. Agricultural Systems, 2002, 71:59-73.
Line 713 The first word capital of the journal name in the reference, other small. In addition, some journals in the literature are full names, some are abbreviations, you should be modified in accordance with the requirements of the contribution journals.
Line 228, 237 , Note that there are some places to be spaced
Line 77,271 When a plant species first appears in the text, the Latin name should be full name. It should be short when it comes back later. Pay attention to Dactylis glomerata,Arabidopsis thaliana Line 119,123,261,139
Fig. 7, 11,12 Lack of differentiation significantness analysis.
some of the figs are not high quality.

Validity of the findings

It is OK.

Additional comments

On the whole, the paper writing is relatively rough, there are many small problems in the text, in addition to the literature reading is not enough, did not quote the literature of recent years (2019-2021), English writing is not standardized, there are many problems, it is recommended to find good Experts in English to promote.

Annotated reviews are not available for download in order to protect the identity of reviewers who chose to remain anonymous.

Reviewer 2 ·

Basic reporting

No comment.

Experimental design

The experiments of abiotic stress(heat,drought,submergence) are all from the literature. Therefore, they don't have to be that specific.

Validity of the findings

No comment.

Additional comments

1.The writing needs polishing.
2.All descriptions of abiotic stresses should be refined by reference.

---

## Round 0.2 · Major Revisions

The authors should carefully consider the recommendations by the reviewers for their resubmission.

Reviewer 1 ·

Basic reporting

See general comments.

Experimental design

Did the authors studied the gene location?
Did the authors perform a genetic transformation and functional analysis of the gene(s) you cloned?
no

Validity of the findings

no

Additional comments

This study was carried out WRKY cloning and analysis of genesin Dactylis glomerata. 98 DgWRKY genes and 281 RGAs,GO and KEGG enrichment analysis of all genes showed that 78 DgWRKY TFs were identified in the plant–pathogen interaction pathway, with 59 of them differently expressed.
In fact, these genes studies have a lot of reports in recent years (about 800 papers in recent three years). Please see the following references. By studying this papers we can recommend some papers to let you know what has happened in recent years. You also can show some progress.
[1] Wang, X. Y., Song, J., Xing, J. H., Liang, J. F., & Ke, B. Y. (2020). Genome-Wide Identification and Expression Profile Analysis of WRKY Family Genes in Teak (Tectona Grandis).
[2] Yang, X., Zhou, Z., Fu, M., Han, M., & Xu, F. (2020). Transcriptome-wide identification of wrky family genes and their expression profiling toward salicylic acid in Camellia japonica. Plant Signaling & Behavior, 16(1), 1844508.
[3] Aguayo, P., Lagos, C., Conejera, D., Medina, D., & Valenzuela, S. (2019). Transcriptome-wide identification of wrky family genes and their expression under cold acclimation in Eucalyptus globulus. Trees, 33(10), 1-15.
[4] Villano, C., Esposito, S., D'Amelia, V., Garramone, R., & Carputo, D. (2020). Wrky genes family study reveals tissue-specific and stress-responsive tfs in wild potato species. Scientific Reports, 10(1).
[5] Shi, Y., Wang, G., Yang, X., & Cao, F. (2019). Analysis of codon usage bias of gene factors in Ginkgo biloba WRKY family. Molecular Plant Breeding.
[6] Wang, X., Li, J., Guo, X., Ma, Y., & Guo, J. (2019). Plwrky13: a transcription factor involved in abiotic and biotic stress responses in Paeonia lactiflora. International Journal of Molecular Sciences, 20(23), 5953.
[7] Wu, G. Q., Li, Z. Q., Cao, H., & Wang, J. L. (2019). Distributed under creative commons cc-by 4.0 genome-wide identification and expression analysis of the wrky genes in sugar beet (Beta vulgaris l.) under alkaline stress. PeerJ, 7(11).
[8] Yan, H., Li, M., Xiong, Y., Wu, J., & Ma, G. (2019). Genome-wide characterization, expression profile analysis of wrky family genes in Santalum album and functional identification of their role in abiotic stress. International Journal of Molecular Sciences, 20(22), 5676-.
[9] Li, Z., X Hua, Zhong, W., Yuan, Y., Wang, Y., & Wang, Z. , et al. (2019). Genome-wide identification and expression profile analysis of WRKY family genes in the autopolyploid Saccharum spontaneum. Plant and Cell Physiology (3), 3.



The authors should be revise the manuscript in the following ways:
The manuscript had showed too much about the function and description of the WRKY gene in the introduction section. The author should directly let the reader known what you should do.
Did the authors studied the gene location?
Did the authors perform a genetic transformation and functional analysis of the gene(s) you cloned?

Annotated reviews are not available for download in order to protect the identity of reviewers who chose to remain anonymous.

Reviewer 2 ·

Basic reporting

The literature of recent years (2019-2021) must be quoted, and some old literatures should be deleted.
The introduction requires refinement.

Experimental design

no comment

Validity of the findings

Fig.10 is lack of differentiation significantness analysis between highly resistant PI251814 and highly susceptible PI292589 lines, between treatment group and control group. And by now, the result can not verify if the up-regulated and down-regulated of the WRKY genes are truely.
Line346-Line361 should also discribe the result of the differentiation significantness analysis.

---

## Round 0.3 · accepted · Accept

The manuscript has been significantly improved by incorporating the suggestions made by the reviewers.